

# Adaptation of metal and antibiotic resistant traits in novel β-Proteobacterium *Achromobacter xylosoxidans* BHW-15

Arif Istiaq[1,2], Md. Sadikur Rahman Shuvo[1,2], Khondaker Md. Jaminur Rahman[1], Mohammad Anwar Siddique[1], M. Anwar Hossain[1] and Munawar Sultana[1]

[1] Department of Microbiology, University of Dhaka, Dhaka, Bangladesh
[2] Department of Microbiology, Noakhali Science and Technology University, Noakhali, Bangladesh

Corresponding author
Munawar Sultana,
munawar@du.ac.bd

## ABSTRACT

Chromosomal co-existence of metal and antibiotic resistance genes in bacteria offers a new perspective to the bacterial resistance proliferation in contaminated environment. In this study, an arsenotrophic bacterium *Achromobacter xylosoxidans* BHW-15, isolated from Arsenic (As) contaminated tubewell water in the Bogra district of Bangladesh, was analyzed using high throughput Ion Torrent Personal Genome Machine (PGM) complete genome sequencing scheme to reveal its adaptive potentiality. The assembled draft genome of *A. xylosoxidans* BHW-15 was 6.3 Mbp containing 5,782 functional genes, 1,845 pseudo genes, and three incomplete phage signature regions. Comparative genome study suggested the bacterium to be a novel strain of *A. xylosoxidans* showing significant dissimilarity with other relevant strains in metal resistance gene islands. A total of 35 metal resistance genes along with arsenite-oxidizing *aioSXBA*, arsenate reducing *arsRCDAB*, and mercury resistance *merRTPADE* operonic gene cluster and 20 broad range antibiotic resistance genes including β-lactams, aminoglycosides, and multiple multidrug resistance (MDR) efflux gene complex with a tripartite system OM-IM-MFP were found co-existed within the genome. Genomic synteny analysis with reported arsenotrophic bacteria revealed the characteristic genetic organization of *ars* and *mer* operonic genes, rarely described in β-Proteobacteria. A transposon *Tn21* and mobile element protein genes were also detected to the end of *mer* (mercury) operonic genes, possibly a carrier for the gene transposition. In vitro antibiotic susceptibility assay showed a broad range of resistance against antibiotics belonging to β-lactams, aminoglycosides, cephalosporins (1st, 2nd, and 3rd generations), monobactams and even macrolides, some of the resistome determinants were predicted during in silico analysis. KEGG functional orthology analysis revealed the potential of the bacterium to utilize multiple carbon sources including one carbon pool by folate, innate defense mechanism against multiple stress conditions, motility, a proper developed cell signaling and processing unit and secondary metabolism-combination of all exhibiting a robust feature of the cell in multiple stressed conditions. The complete genome of the strain BHW-15 stands as a genetic basis for the evolutionary adaptation of metal and the antibiotic coexistence phenomenon in an aquatic environment.

## INTRODUCTION

With the passage of time, bacteria have acquired a number of mechanisms for both metal and antibiotic resistances upon evolution. The genetic plasticity of bacteria allows them to acquire such survival strategies by mutations, alteration of gene expression or genetic material acquisition which leads to the harborage of resistance determinants within (*Silver & Phung, 1996*; *Munita & Arias, 2016*).

Metals are common elements found throughout the earth's crust naturally, and these are widely distributed in the environment (*Tchounwou et al., 2012*). Each metal maintains a distinct biogeochemical cycle on earth and can transfer from animal to bacteria as part of cycling processes. Some metals are essential and some are toxic to cellular system depending on metal species as well as cell type. For a bacterium, metals including nickel, iron, copper, and zinc are required as trace elements and are essential for some metabolic reactions. On the contrary, some metals such as mercury, silver and cadmium are harmful even at very low concentration and have no biological role to the organism (*Hughes & Poole, 1989*). Bacterial associations with metals are quite diverse and the genomic level induction for their tolerance or transformation depends on the exposure level.

On the other hand, extensive use of many antibiotics and their disposal from clinical waste and industrial origin may contaminate the water. This can act as an inducer for the dispersion of antibiotic resistance prevalence in both clinical and environmental bacteria (*Salyers & Amabile-Cuevas, 1997*; *Walsh, 2006*). Also, it can lead to a potential alteration of microbial ecosystems affecting their community composition and functions (*Baquero, Martínez & Cantón, 2008*).

Bacteria can acquire resistance to both metals and antibiotics simultaneously. The co-selection of these resistances can be caused by co-resistance or cross-resistance occurring through multi-resistance genetic elements such as transposon, integron, and plasmid (*Baker-Austin et al., 2006*). A rising concern is the development of antimicrobial resistance in metal contaminated environments (*Wright et al., 2006*; *Matyar, Kaya & Dincer, 2008*; *Tuckfield & McArthur, 2008*). A possible mechanism could be the selection of metal resistance by metal stress acting as a determinant for the antibiotic resistance acquisition making a bridge between non-antibiotic agents (e.g., metal) and antibiotic resistance (*Summers et al., 1993*). It is possible that the way of the selection of resistance variants for metal may be similar to the selection of antibiotic-resistant strains.

Bangladesh has been noted as the largest massive As poisoning occurrence country by the World Health Organization (WHO). The reason behind the exposure is for naturally occurring inorganic As accumulation in the groundwater of several colonized areas (*Smith, Lingas & Rahman, 2000*). Therefore bacteria colonizing such environment might have some As resistance or conversion mechanism within. Previously we found diverse arsenotrophic bacteria living in arsenic affected groundwater in different areas of

Bangladesh (*Sultana et al., 2017*). Having such a metal resistant microbiome its worth studying their genomic potentiality as well as there other possible resistance mechanisms. Moreover, a study showed that metal could induce antibiotic resistance in bacteria (*Chen et al., 2015*). A genome-wide analysis provides an area for better understanding of possible environmental co-selection and adaptation processes in bacteria against metals and antibiotics along with unique metabolism and survival potential in stressed environment. Therefore, in this study, a draft genome of environmental strain *Achromobacter xylosoxidans* BHW-15 collected from As contaminated ground water is reported and analyzed thoroughly to reveal the genomic features and its innate resistance focusing on multi-metal resistance and multidrug resistance (MDR), core metabolism and adaptation potentiality. The isolate was retrieved from As contaminated tubewell water (total As content 0.01 mg/L) collected from the Bogra district of Bangladesh.

## MATERIALS AND METHODS

### Bacterial isolation and screening of arsenite transformation

The isolate designated as BHW-15 was retrieved from a tubewell water in Bogra District of Bangladesh on arsenite supplemented heterotrophic growth medium (*Sultana et al., 2017*). Arsenite transformation potential was analyzed by both $KMnO_4$ and $AgNO_3$ assay. Qualitative $KMnO_4$ screening method was used to determine the arsenite conversion initially (*Fan et al., 2008*). $KMnO_4$ has characteristic pink color and it is a highly oxidizing agent. A total of 500 µL culture was taken in 1.5 mL micro-centrifuge tube and 10 µL of 0.05M $KMnO_4$ was added and the color change was monitored. Phenotypic $KMnO_4$ was verified by $AgNO_3$ test (*Salmassi et al., 2002*). The isolate was streaked on heterotrophic solid medium containing two mM sodium arsenite and incubated at 30 °C. After the growth, 0.1 M of $AgNO_3$ solution was added to the growth plate. Formation of a brown precipitate was observed. The isolate was also analyzed for the presence of arsenite-oxidizing a*ioA* gene by polymerase chain reaction using specific primers (*Quemeneur et al., 2008*) (Forward: 5′-CCACTTCTGCATGCTGGGMTGYGGNTA-3′, Reverse: 5′- TGTCGTTGCCCCAGATGADNCCYTTYT-3′) followed by Sanger sequencing of the PCR product to confirm the gene sequence.

### Genome sequencing and assembly

DNA from the pure culture of the isolate BHW-15 was extracted using QIAamp DNA Mini Kit (Qiagen, Hilden, Germany) according to the manufacturer's instructions. The quality and quantity of the extracted genomic DNA were assured by Nanodrop ND-200 (Thermo Fisher, Waltham, MA, USA) and the integrity was assured by agarose gel electrophoresis. Whole genome sequencing was performed by Ion-Torrent High Throughput Sequencing technology. Machine generated data was transferred to the Ion Torrent server where data was processed through signal processing, base calling algorithms and adapter trimming to produce mate pair reads in FASTQ format. The FASTQ reads quality was assessed by the FastQC tool (*Andrews, 2010*) followed by trimming of low quality reads and reads less than 200 bp using the Trimmomatic tool (*Bolger, Lohse & Usadel, 2014*), where quality cut off value was Phred-20. De novo assembly of the reads was

performed using SPAdes, (version 3.5.0) genome assembler (*Bankevich et al., 2012*). Generated assembled reads were mapped and reordered according to a reference sequence of *A. xylosoxidans* A8 complete genome from NCBI (accession number: NC_014640.1) by progressive Mauve algorithm in Mauve software (*Darling et al., 2004*).

## Identification of bacterial species

Assembled contigs were analyzed by BLAST and the k-mer algorithm in the KmerFinder 2.0 tool to identify the bacterium at species level (*Hasman et al., 2014*; *Larsen et al., 2014*). Whole genome based phylogenetic analysis was performed using REALPHY (*Bertels et al., 2014*). Annotated genome comparison was performed by locally collinear block method in Mauve (*Darling et al., 2004*). The Plasmid Finder 1.3 tool was used for the detection of plasmid sequence contamination (*Carattoli et al., 2014*).

## Genome annotation, analysis and metabolic reconstruction

The assembled draft genome of BHW-15 was annotated by multiple annotation schemes to improve accuracy. Used software includes NCBI Prokaryotic Genome Annotation Pipeline (*Tatusova et al., 2016*), PROKKA ($e = 0.000001$) (*Seemann, 2014*), RAST ($e = 0.000001$) (*Aziz et al., 2008*) and KAAS (*Moriya et al., 2007*). Annotated genes by each software were then cross checked for each tool. For the detection of tRNA and tmRNA, tRNAscan (*Lowe & Eddy, 1997*) and Aragorn (*Laslett & Canback, 2004*) software were used accordingly. Secondary metabolite gene clusters were identified by anti-SMASH version 4.0.2 software (*Medema et al., 2011*). The SEED viewer (*Aziz et al., 2012*) was used for the exploration and comparative analysis of annotated genes. KEGG MGMapper tool (*Kanehisa & Goto, 2000*) was used for metabolic pathway reconstruction.

## Antibiotic susceptibility assay

Antibiotic susceptibility test was conducted by Kirby–Bauer disk diffusion method (*Bauer et al., 1966*). A total of 14 antibiotics belonging to 10 antibiotic groups covering six different mode of action including oxacillin, ampicillin, cefalexin, cefuroxime, cefotaxime, cefepime, aztreonam, polymixin B, gentamicin, doxycycline, chloramphenicol, azithromycin, nalidixic acid, and nitrofurantoin were used for this study. Antibiotic susceptibility was interpreted referring to the CLSI guidelines and Antimicrobe Database (*Roberts & Lang, 2009*; *Clinical and Laboratory Standards Institute, 2009*).

## Accession number

The whole genome datasets generated and analyzed during the current study are available in the NCBI GENBANK repository (accession number: PZMK00000000.1).

## RESULTS

Whole genome sequencing of the arsenite oxidizing isolate BHW-15 identified the bacterium as *A. xylosoxidans*. The assembled filtered draft genome of *A. xylosoxidans* BHW-15 strain was 6,301,677 bp assembled into 2,049 contigs. The GC content of the genome was 65%. The genome possessed 8,159 Coding sequences (CDS) by RAST, 7,627 by NCBI, and 6,732 CDS by PROKKA. Using RASTtk (*Brettin et al., 2015*) annotation

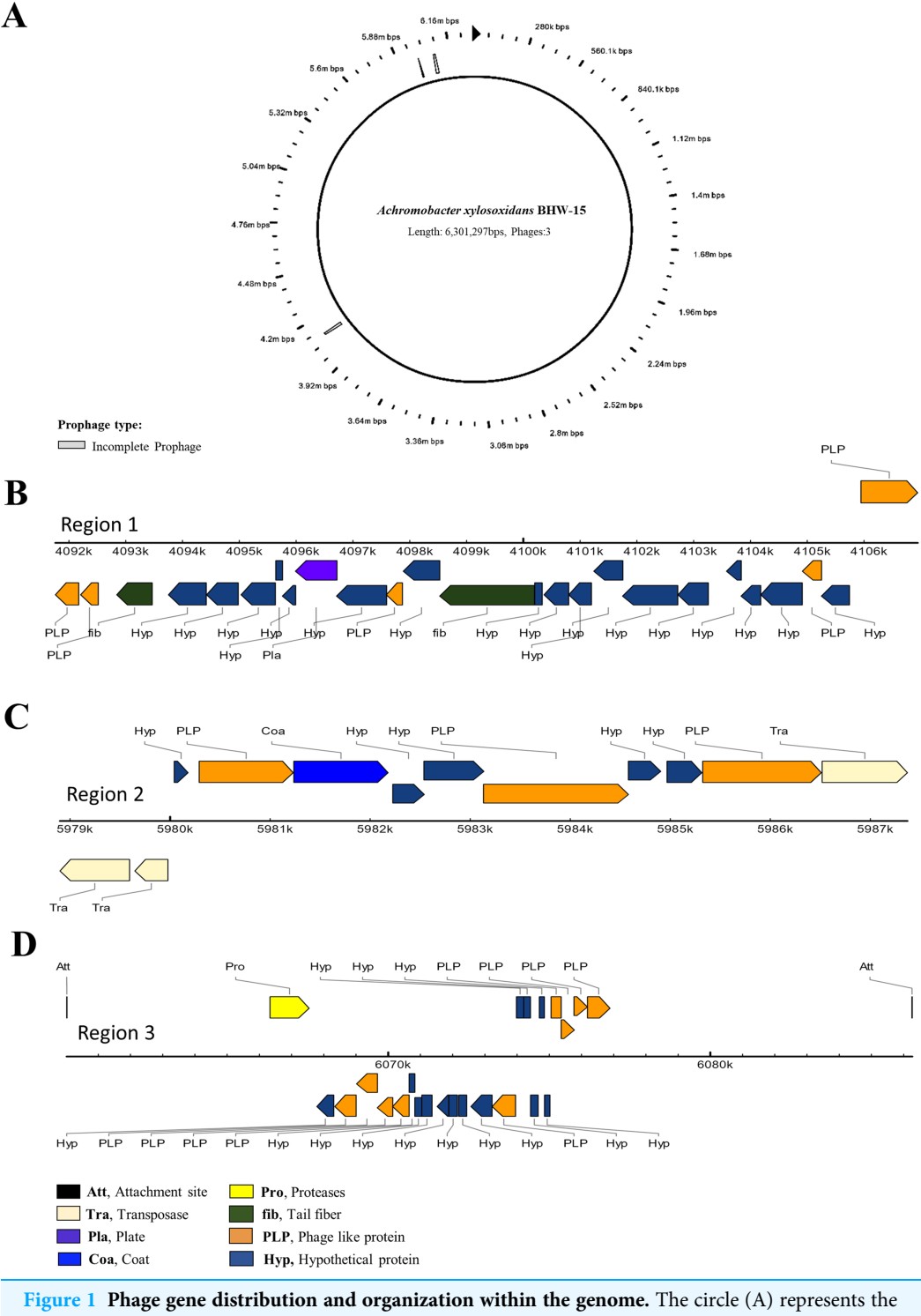

**Figure 1 Phage gene distribution and organization within the genome.** The circle (A) represents the concatenated whole reordered genome. In linear representation of regions (B–D), colored shapes indicate the relative position and type of the genes it contains.

scheme in the RAST software, 32% of all genes were located in the generated subsystems and the rest 50% were out of the subsystem list. The genome contained three incomplete phage site containing phage-associated phage body protein genes indicating
**Table 1 General features of the *Achromobacter xylosoxidans* BHW-15 genome.**

| Feature | Value |
| --- | --- |
| Draft genome Size | 6,306,677 |
| GC content | 65% |
| NCBI (GeneMarkS+) | |
| Number of coding sequences (CDS) | 7,627 |
| Number of protein coding genes | 5,782 |
| Number of pseudo genes | 1,847 |
| Number of RNAs | 125 |
| Number of tRNAs | 64 |
| Number of rRNAs | 7,18,32 (5S,16S,23S) |
| Number of ncRNAs | 4 |
| RAST SOFTWARE (RASTtk) | |
| Genes in subsystem | 2,533 |
| Hypothetical | 99 |
| Non-Hypothetical | 2,434 |
| Genes not in subsystem | 5,626 |
| Hypothetical gene | 2,102 |
| Non-hypothetical gene | 3,524 |

multiple phage confrontation (Fig. 1). The basic features of the genome are summarized in Table 1. Annotation results from subsystem and pathway reconstruction are depicted in Fig. 2. BLAST and the SEED close strain analysis found that the genome had similarity with *Bordetella* along with other *Achromobacter* strains. Kmer based genomic comparison and whole genome based phylogeny showed the *A. xylosoxidans* BHW-15 had the closest proximity to the strain "A8" (Fig. 3A). Further functional gene comparison between A8 and BHW-15 using SEED (File S1) and localized co-linear block revealed that the BHW-15 strain differs in metal resistance gene profile to A8 strain. BHW-15 possess a unique genetic island and organization of *aio* resistance island along with distinctive *ars* island that was absent in A8 strain (Fig. 3B).

## Metal resistance genes and operon clusters within the genome of isolate BHW-15

Both metal and antibiotic-resistant genes were present within the genome of BHW-15. A total of 35 metal resistance genes along with two arsenic operon gene clusters (arsenite oxidizing *aioBA* and arsenate reducing *arsRCDAB*) and a mercury resistance *merRTPCADE* operon gene cluster were present where *mer* operon is at the right end of *aio* operon gene cluster. There is a tn21 transposon-like gene to the right end of *mer* operon gene cluster. The genome also possessed copper, zinc, and cadmium resistance-associated genes (Table 2).

*Achromobacter xylosoxidans* BHW-15 harbored two oxidase genes namely *aioA* (arsenite oxidase large subunit and *aioB* (arsenite oxidase small subunit) preceded by a phosphate transporter or inorganic arsenic binding, *aioX* (periplasmic component gene),

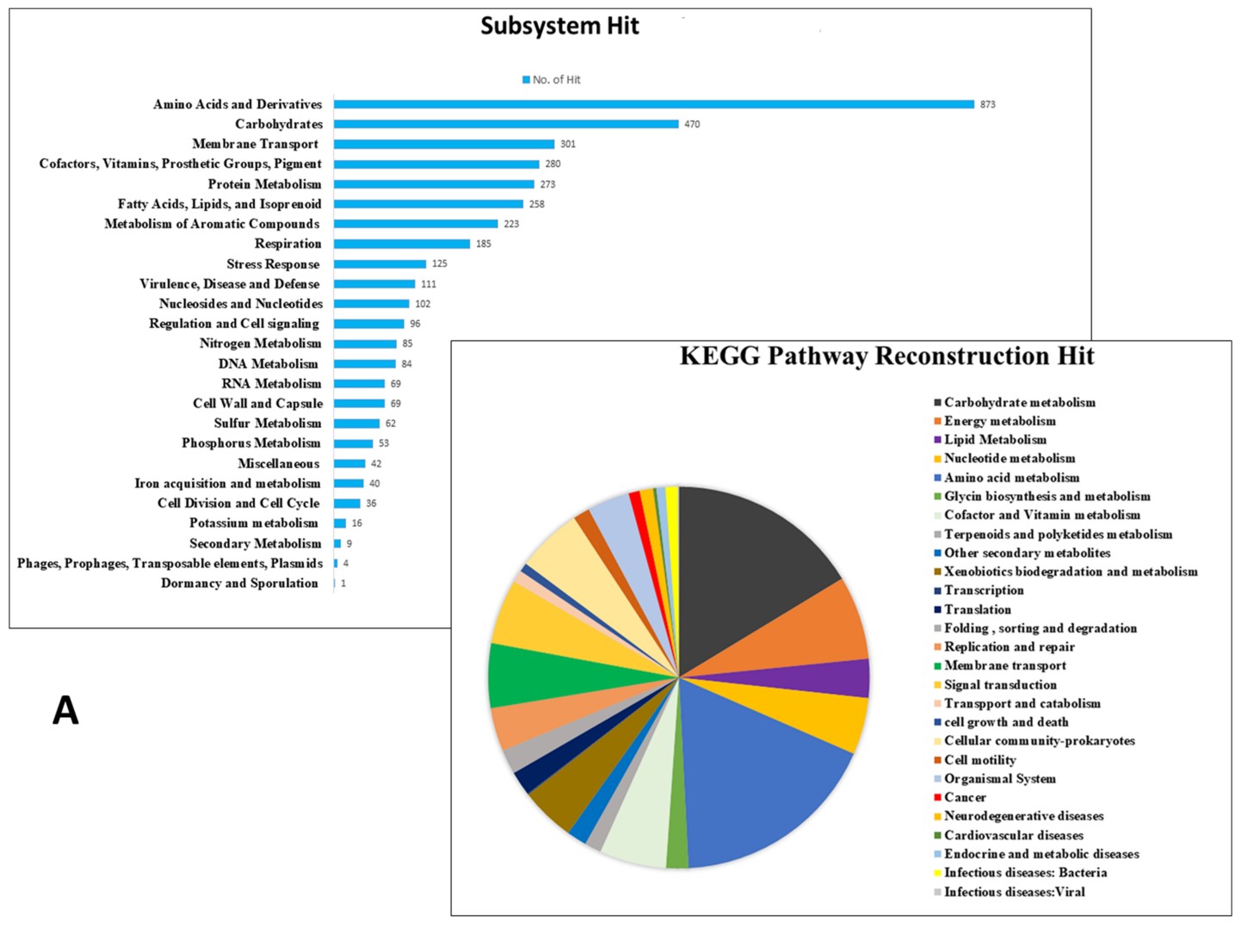

**Figure 2  Subsystem (A) and Pathway reconstruction KEGG (B) hit distribution for *A. xylosoxidans* BHW-15.**

and a sensor histidine kinase, *aioS* (a transmembrane signal transduction gene) (*Wolanin, Thomason & Stock, 2002*). Following *aioA*, there was an electron transporting Cytochrome c551/c552 gene, a Molybdenum cofactor biosynthesis gene *moaA* and an As operon repressor gene. In the island, two version of *aioA* gene were detected in two overlapping reading frames. The genome also contained a complete mercury resistance operon *merRTPADE* gene island near to the arsenite oxidizing *aio* operonic gene island. The distance between these two islands was 2,304 bp with only one gene (putative phosphatase) in-between.

The chromosome of *A. xylosoxidans* BHW-15 also possessed Arsenate reducing operon gene island *arsRCDAB* surrounded by ABC transporter and Phosphate transport

**A**

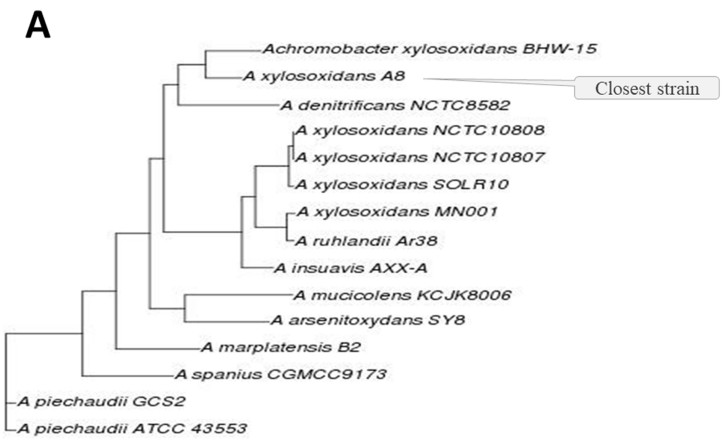

**B**

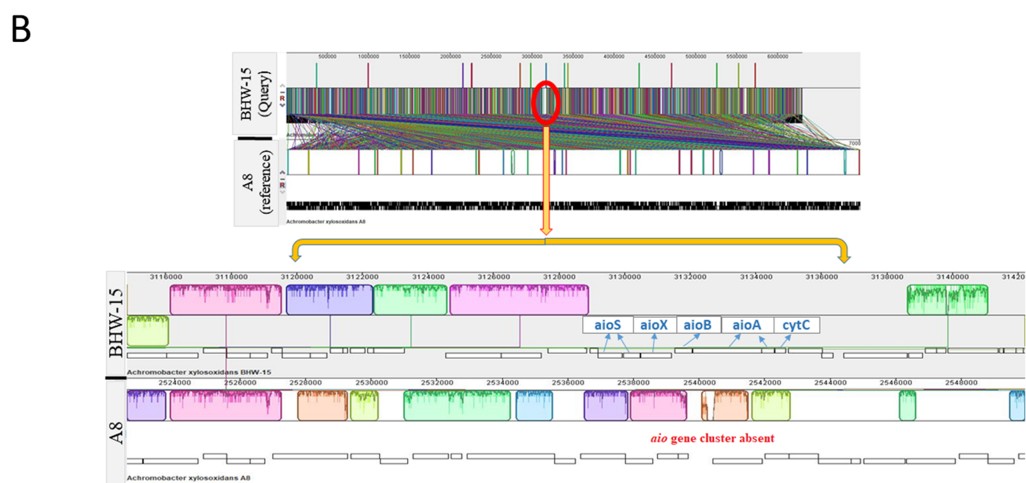

**Figure 3 Genome comparison of closely related *Achromobacter* species.** Whole genome based phylogeny (A) and Locally collinear block (LCB) method (B).

**Table 2 Identified metal resistance genes within the draft genome of *A. xylosoxidans* BHW-15.**

| Metal resistance genes | Genes |
|---|---|
| Arsenite oxidation | *aioS, aioX, aioB, aioA* |
| Arsenate reduction | *arsH, arsR, arsC, arsD, arsA, arsB* |
| Mercury resistance operon | *merR, merT, merP, merC, merA, merD, merE* |
| Mercuric reductase | *miR* |
| Chromium resistance | *chrB, chrA, chrF* |
| Copper homeostasis | *ciA, copZ, clfA, mO, hL, copG, crB* |
| Copper tolerance | *cutA, corC* |
| Zinc resistance | *zraR* |
| Cobalt-zinc-cadmium resistance | *czcA, czcC, czsB, cusA, czrR, hmhK, tR* |

system genes. Synteny analysis with other metal converting bacteria showed that the genomic organization of these resistance island has significant distinction and content similarity with other metal resistant bacteria (Fig. 4).

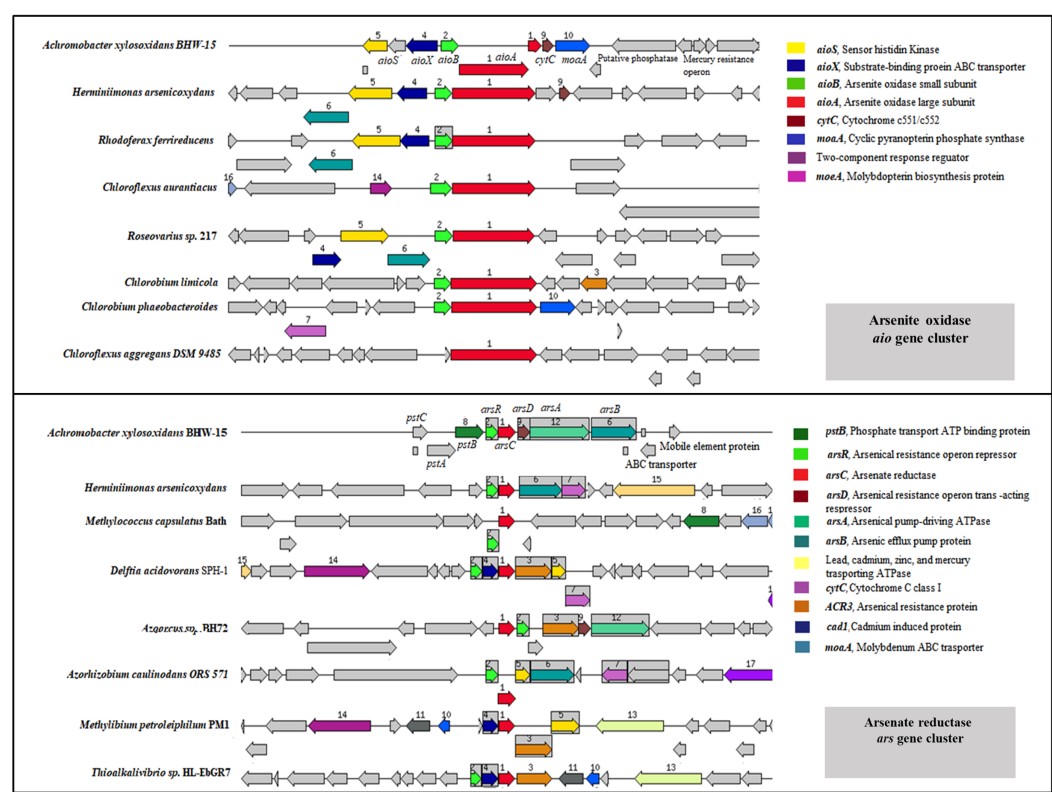

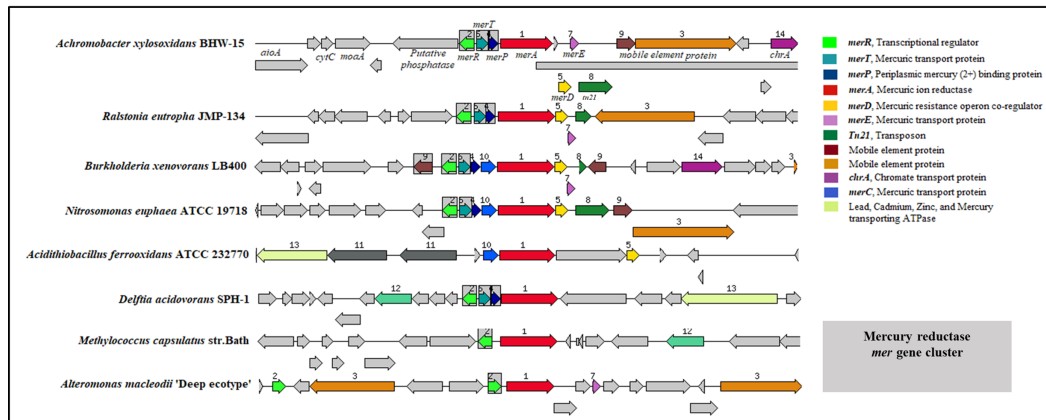

**Figure 4 Synteny analysis and gene organization of Metal resistance operonic gene clusters.**
(A) Arsenite oxidase *aio* and Arsenate reductase *ars* operonic gene clusters. Two red arrows in the *aio* operon of BHW-15 indicate two *aioA* genes in two different overlapping reading frames. (B) Mercury resistance *mer* operonic gene cluster.

## Antibiotic resistance genes within the genome of isolate BHW-15

In *A. xylosoxodans* BHW-15, several antibiotic-resistant genes co-existed along with metal resistance genes. Present genes were involved in several different antibiotic resistance mechanisms by enzymatic degradation and efflux pump systems. According to RAST, the genome possessed resistance genes against β-lactams, Fluoroquinolones, and

**Table 3 Identified antibiotic resistance genes within the draft genome of *A. xylosoxidans* BHW-15.**

| Antibiotic resistance genes | Genes |
|---|---|
| β-Lactamases | |
| β lactamase | *bl* |
| β-lactamase Class C and other penicillin binding proteins | *blC* |
| Aminoglycoside modefying enzyme | |
| Streptomycin 3″-O-adenylyltransferase | *adaA1* |
| Spectinomycin 9-O-adenylyltransferase | *adaA2* |
| Efflux pumps | |
| Outer membrane component of tripartite multidrug resistance system | *OM* |
| Inner membrane component of tripartite multidrug resistance system | *IM* |
| Membrane fusion component of tripartite multidrug resistance system | *MFP* |
| RND efflux system, membrane fusion protein | *cmeA* |
| RND efflux system, inner membrane transporter | *cmeB* |
| RND efflux system, outer membrane lipoprotein | *cmeaC* |
| Transcription repressor of multidrug efflux pump acrAB operon | *acrR* |
| Transcription regulator of multidrug efflux pump operon, of AcrR family | *tetR* |
| Probable transcription regulator protein of MDR efflux pump cluster | *reg* |
| Multi antimicrobial extrusion protein | *mat_all* |
| Multidrug and toxin extrusion (MATE) family efflux pump | *ydhE/norM* |
| Type I secretion outer membrane protein, TolC precursor | *tolC_14* |
| Macrolide-specific efflux protein | *macA,* |
| Macrolide export ATP-binding/permease protein | *macB,* |
| Acriflavin resistance protein | *acrB* |
| MAR locus | |
| Multiple antibiotic resistance protein | *marC* |

Aminoglycosides antibiotics (Table 3). Using in vitro antimicrobial susceptibility tests, the bacteria showed resistance showed resistance against beta lactams from narrow spectrum to broad-spectrum penicillin, 1st generation to 3rd generation cephalosporins, aminoglycosides, monobactam, and macrolides, whereas it showed susceptibility to polymyxin B, tetracycline, nitrofuran and moderate susceptibility to 4th generation ciprofloxacin, chloramphenicol, and nalidixic acid (Table 4). These resistances suggest the possible expression of detected antibiotic resistance genes within the bacterium. Several genes for efflux systems like MDR MAR locus, multidrug-resistant tripartite system and efflux system complex *OM-IM-MFP* were also detected which might facilitates active efflux of antibiotic or metal. Within the genome one pili gene cluster associated with bacterial movement was also present. In addition to the metal and antibiotic resistance, these bacteria possess several genes that may be associated with its defense mechanism and bacterial pathogenesis.

## Signaling and stress response genes

### Metabolic pathway analysis

Pathway reconstruction for the genes of *A. xylosoxidans* BHW-15 genome produces a number of complete and incomplete pathways (one or two blocks missing). In the genome

**Table 4 Antimicrobial susceptibility test toward different antibiotics for *A. xylosoxidans* BHW-15.**

| Antibiotic group | Mode of action | Generic name | Antibiotic | Result |
|---|---|---|---|---|
| Penicillins | B-lactamase inhibit | Oxacillin (narrow) | OX1 | R |
| | | Ampicillin (broad) | AMP10 | R |
| Cephalosporin (1st G) | | Cefalexin | CL 30 | R |
| Cephalosporin (2nd G) | | Cefuroxime | CXM 30 | R |
| Cephalosporin (3rd G) | | Cefotaxim | CTX 30 | R |
| Cephalosporin (4th G) | | Cefepime | FEP 30 | MS |
| Monobactams | | Aztreonam | ATM 30 | R |
| Polymixin B | Outer membrane permeability | Polymixin B | Pb300 | S |
| Aminoglycosides | Protein synthesis (30S) | Gentamicin | Cn10 | R |
| Tetracycline | | Doxycycline | DO 30 | S |
| Chloramphenicol | Protein synthesis (50S) | Chloramphenicol | C 30 | MS |
| Macrolides | | Azithromycin | AZM 15 | R |
| Quinolone | DNA Topoisomerase | Nalidixic acid | NA 30 | MS |
| Nitrofurans | DNA damage | Nitrofurantoin | F 300 | S |

**Note:**
R, Resistant; S, Sensitive; MS, Moderately sensitive; G, Generation.

these detected pathways are distributed under five major KEGG pathway categories including cellular process, metabolism, environmental information processing, genetic information processing, and human diseases (Fig. 2). BHW-15 possesses several genes that are associated with different type of adaptation supporting pathways that might help the bacteria being robust in its life span. These pathways include motility, flagellar assembly, quorum sensing, biofilm formation; biosynthesis of vitamin, co-factors, folate, one carbon pool, secondary metabolites like terpinoids, polyketides etc. Moreover, several xenobiotic degradation metabolism pathways like amino benzoate degradation, cytochrome P450 were also found supported according to the genomic context.

### Secondary metabolites analysis

According to AntiSMASH, the genome possesseses five secondary metabolite gene clusters with 14 putative gene clusters. Detected secondary metabolites include ectoin (osmolites protective substance) (*Bernard et al., 1983*), resorcinal, arylopolyene (protects from reactive oxygen species) (*Schöner et al., 2016*), and terpene (ecological role), phosphonate (global phosphorus cycle) (*Yu et al., 2013*) that supports the bacteria for surviving in harsh environment (File S2).

## DISCUSSION

The genome analysis helps to achieve a pertinent inference for the coexistence of metal and antibiotic resistance genes. The isolate was retrieved from an arsenic contaminated tubewell water. Therefore, it can be expected to find some sort of As resistance in the genome. In genome analysis, two As metabolizing operon like gene clusters *aioSXBA* (oxidation) and *arsRCDAB* (reduction) were detected. Presence of such dual system was previously reported in highly efficient arsenite oxidizing bacteria *Achromobacter*
*arsenitoxydans* SY8 (*Li et al., 2012*) suggesting the *A. xylosoxidans* BHW-15 could also have a high efficiency in arsenite oxidation. Analysis of synteny found that such organization of *aio* was quite distinctive. A possible way for such organization to function in arsenite oxidation is, AioS sense the As (III) in the environment generating proper signal to activate other genes, AioX binds and transport the arsenite into the cell, *aioBA* encodes the oxidase enzyme which converts arsenite to arsenate, CytC accepts the electron and transport to the cellular electron transport chain and finally the system is controlled by the arsenical resistance repressor gene in the right end of the operon. This scenario supports the aerobic respiratory oxidation of arsenite (*Gihring & Banfield, 2001*). In arsenate reduction by *ars* genes, *arsC* encodes the arsenate reductase and expand substrate specificity for transporting by efflux transporters ArsB, ArsR, and ArsD function as primary and secondary regulators of the *ars* operon accordingly, and ArsA encodes membrane-associated ATPase protein attached to ArsB energizing the efflux pump by ATP hydrolysis (*Rosen, 1990*; *Liu & Rosen, 1997*; *Silver, 1998*; *Li et al., 2002*). Such arrangement of arsenate reductase was similar to other high arsenate metabolizing bacteria. Notably, both of the *aio* and *ars* operons were similar to high arsenic transforming *Herminimonas* sp. While *Herminimonas* sp. is rare in the environment and found in highly metal contaminated zone, BHW-15 was found in groundwater but gained similar genes, indication of high arsenic resistance activity. However, how these two As oxidation-reduction systems are regulated in the bacterial cell in an aquatic environment is not clearly understood. Considering the functions and the genetic organization a possible mechanism for the regulation of these two systems can be deduced. In which, both operons can either work individually and efflux out converted arsenate (*aio*) and arsenite (*ars*) or work as a unit where environmental arsenite is converted to arsenate inside the cell by *aio* operonic genes and then re-converted upon expedition of deposition tolerance, to arsenite (ArsC), leading to the eviction (ArsB) from the cell by *ars* operon. Thus, detoxification of arsenite and arsenate is performed (*Carlin et al., 1995*). However, *aio* operonic island didn't have any regulator of its own; therefore, these cluster might work with other functional operon to perform successfully which could be the *mer* operon existed just upstream to *aio* operon or may be with *ars* operon as suggested above. This mercury resistance *mer* operon was found juxtaposed to *aio* (2,304 bp gap with a putative phosphatase gene in between). At the end of *mer* operon, a tn21 gene was found suggesting a transposon-mediated resistance development in the bacterial genome (*Cynthia, Ruth & Anne, 1999*). In *mer* operon, *merR* and *merD* act as regulator and coregulator consecutively, *merP* encodes for periplasmic mercury (2+) binding protein, *merA* encodes for mercuric ion reductase, and *merT, merC,* and *merE* encode for mercury transporting protein (*Nascimento & Chartone-Souza, 2003*). Along with As and Hg, the genome also possesses resistance genes against Cu, Zn, Cd, Cr, and Co. Metals can interfere with several bacterial cellular functions like protein activity, oxidation, nutrient assimilation, membrane stability, and DNA replication (*Nies, 1999*; *Lemire, Harrison & Turner, 2013*). In response, bacteria develop metal resistance by efflux, reduction, detoxification, or biofilm formation (*Harrison et al., 2008*). Although, Cu and Zn are essential as trace elements others are very toxic to bacteria (*Hughes & Poole,*

*1989*). Detected Cu, Zn, Cd, and Cr resistance genes were found involved in similar functions for resistance mechanism such as sensing (*cusR, czrR, hmhK*), efflux (*ciA, clfA, czcA, cusB, chrA, corC*), oxidation (*mO*), resistance (crB, *czcA, czcD, chrB*), and repressor (*copG, tR, cusR, zraR*).

In a similar manner, the genome harbors several antibiotic resistance genes responsible for beta-lactamase, aminoglycosides, MDR efflux pumps, and MDR tripartite system. Detected *bL* and *blC* genes encode for beta-lactamase and other penicillin binding protein protecting against beta-lactam ring inhibiting antibiotics (*Neu, 1969*). Also, in supportive manner, pathway analysis showed that these genes support a complete beta-lactam resistance pathway. In the antibiotic susceptibility test, the isolate showed resistance against almost all beta-lactams such as penicillin, 1st–3rd generation cephalosporin, monobactam, and moderately sensitive to the 4th generation cephalosporin. The isolate also found resistant to aminoglycosides with *adaA1* and *adaA2* adenyl transferase genes protecting against protein synthesis inhibition, along with moderately sensitive to chloramphenicol and quinolone. Thus, *A. xylosoxidans* BHW-15 has developed a multilayer resistance to membrane modification to protein-DNA synthesis inhibition by various antibiotic drugs. With some deviation these antibiotic resistance genes were also found in other *A. xylosoxidans* strains (*Amoureux et al., 2013*; *Jakobsen et al., 2013*).

Therefore, all these antimicrobial resistance genes might have been active within the genome that satisfy the explanation for such broad range antimicrobial resistance activity showed by the isolate. This broad metal and antibiotic resistance scenario indicates the evolutionary adaptation and resistance development of the bacteria by high selective pressure in the environment. In the genome, presence of pili genes for motility and stress tolerating genes for heat-cold shock, detoxification, osmotic pressure, carbon starvation, dormancy, etc. support the bacterial efficacy for thriving in high contamination (*Stevenson, 1977*; *Thieringer, Jones & Inouye, 1998*; *Watson, Clements & Foster, 1998*; *Mille, Beney & Gervais, 2005*; *Jones & Lennon, 2010*; *Maleki et al., 2016*; *Essa, Al Abboud & Khatib, 2018*; *Sun, Liu & Hancock, 2018*).

In addition, three phage signatures in the chromosome also indicate bacterial adaptation and recovery against phage attack. Phage propagation in the environment can determine the host bacterial diversity and variation (*Casjens, 2003*; *Koskella & Meaden, 2013*; *Parmar et al., 2017*).

These three phage regions (Fig. 1) within the genome can be explained as the survival of bacteria which also leads to the bacterial robust feature (*Koskella & Brockhurst, 2014*). It is not known if these phage immunities have some effect with such high metal—antibiotic resistance or vice-verse but it shows a possibility that this resistance potential might somehow facilitate the bacteria to overcome phage attacks.

Hence, considering the co-existence phenomenon, some questions arise like how this bacterium could have achieved such antibiotic and metal cross resistances in the natural aquatic environment, how these two different resistance might interact with each other and is there any impact of one resistance to the regulation of the other one. A whole genome study of pathogenic strain *A. xylosoxidans* NH44784-1996 causing cystic fibrosis revealed almost similar metal-antibiotic resistance genes possession. But there was

a significant difference in the metal and antibiotic resistome. Such as arsenite oxidizing genes in BHW-15 were completely absent in NH44784-1996 and antibiotic resistance genes and developed resistance in NH44784-1996 was much higher than BHW-15 strain (Tables 2 and 3) (*Jakobsen et al., 2013*).

Therefore, a possible explanation for this co-existence might be, in the environment metal exerts a selective pressure that indirectly induces the selection of antibiotic resistance, especially in the environment contaminated with these two elements (*Foster, 1983*; *McIntosh et al., 2008*) or alternatively antibiotics might exert positive selective pressure on bacteria to acquire metal resistance, thus one resistance might have promoted the development of others leading to the coexistence phenomenon in bacteria (*Baker-Austin et al., 2006*).

While in the aquatic environment, antibiotics from the industrial origin and antibiotic resistance genes in pathogenic bacteria from animal origin are circulating due to excessive use of the antibiotics (*Walsh, 2006*). It has been reported that antibiotic disturbance in the environment affects primary microbial process such as nitrogen transformation, methanogenesis, and sulfate reduction (*Ding & He, 2010*). Likewise, as antibiotic interferes with the fundamental cellular process it may also impact on metal resistance gene regulation. Alternatively, the metal may also interfere with antibiotic resistance gene regulation. Therefore, the presence of antibiotic resistance genes might have impact on the metal resistance regulation or the vise verse. In the KEGG analysis, a number of cellular processes like survival and adaptation promoting pathways have been observed within the genome. It can be inferred from its genetic potential of quite developed carbon and energy metabolisms that these systems might facilitate the bacterium to utilize diverse carbon source for energy production and surviving diverse contaminated environment. Presence of the multiple stress tolerant genes and genes for defensive efflux pump, signaling, quorum sensing, flagella, anti-toxin metabolism all together gives an idea that total process in combination might be helping the bacteria to go on through resistance adaptation process. In genome comparison some arsenic related genes were not observed in the same species from different source. This indicates the existence of resistant determinant acquisition mechanism of the bacterium upon necessity. Also, genomic profile suggests a possible mobilome of metal resistance in the bacteria (As, Hg) where there was a Tn21, the flagship of transposon on the upstream of the As-Hg operons. Disease association based on the resistance genes that were found is worth to study further (*Harbottle et al., 2006*; *Jakobsen et al., 2013*; *Ventola, 2015*; *Zaman et al., 2017*). Described association also supports the bacterial pathogenicity as an opportunistic pathogen indicating it might exacerbate the disease condition of immune suppressed patient (*Newman et al., 1984*; *Turel et al., 2013*; *Dupont et al., 2018*).

All together the total scenario is alarming to consider the multi-potential ability of this environmental *A. xylosoxidans* BHW-15 and cast a new insight on metalantibiotic resistance proliferation. Further study of the other arsenic or metal metabolizing bacteria focusing on antibiotic metal cross resistance might reveal the inner mechanism and the future resistance pattern and risk to the environment.

## CONCLUSIONS

Finally, our data supports the hypothesis that environmental selective pressure of antibiotic or metal from pollution can lead to the development of multi-metal and antibiotic-resistant bacteria. It also establishes a possibility for the interaction between the metal and antibiotic resistance regulation and metabolic potentiality in relation. Thus, it stands as a basis for further co-existence of resistance and metabolic potential interaction study to understand the metal-antibiotic resistance interaction in biogeochemical cycle and its impact on microbiome.

## ACKNOWLEDGEMENTS

The corresponding author thanks Ms. Farzana Diba, Scientific officer in the Bangladesh Atomic Energy Commission and PhD student at the Department of Microbiology, University of Dhaka for her assistance in bacterial isolation.

### Funding

This work was supported by the University Grants Commission (UGC) and Ministry of Science and Technology, Govt. of Bangladesh. The funders had no role in study design, data collection and analysis, decision to publish, or preparation of the manuscript.

### Grant Disclosures

The following grant information was disclosed by the authors:
University Grants Commission (UGC).
Ministry of Science and Technology.

### Competing Interests

The authors declare that they have no competing interests.

### Author Contributions

- Arif Istiaq performed the experiments, analyzed the data, prepared figures and/or tables, approved the final draft.
- Md. Sadikur Rahman Shuvo performed the experiments, analyzed the data, approved the final draft.
- Khondaker Md. Jaminur Rahman performed the experiments, approved the final draft.
- Mohammad Anwar Siddique analyzed the data, approved the final draft.
- M. Anwar Hossain authored or reviewed drafts of the paper, approved the final draft, dr. Hossain provided laboratory facilities and guidance.
- Munawar Sultana conceived and designed the experiments, contributed reagents/materials/analysis tools, authored or reviewed drafts of the paper, approved the final draft, dr. Sultana managed the grant under which the research was conducted.

## Data Availability

Data is available at the NCBI GenBank repository, accession number PZMK00000000.1.

## Supplemental Information

Supplemental information for this article can be found online at http://dx.doi.org/10.7717/peerj.6537#supplemental-information.

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
