# Peer review of "Adaptation of metal and antibiotic resistant traits in novel β-Proteobacterium Achromobacter xylosoxidans BHW-15"

_PeerJ, doi:10.7717/peerj.6537_

## Round 0.1 · original submission · Major Revisions

Two of 3 reviewers had concerns about the grammar and presentation overall. I would suggest that once you made all the technical revisions to your manuscript that you have it edited for grammar by a native English speaker. Proper grammar and presentation will be important criteria I will use to determine if your revised manuscript is ultimately acceptable.

There were also technical comments that need to be addressed in your revision. In some instances, it appears that some important experimental details were missing.

Reviewer 1 ·

Basic reporting

The manuscript is well prepared, language is fine, methods are well presented and results are discussed and from my point of view conclusions are justified.

Experimental design

Well conducted and reported.

Validity of the findings

Results are presented and discussed properly. It seems to me robust.

Additional comments

I only have minor comments which should be addressed and hopefully improve the manuscript.
Overall, this work will gain attention by specialist but also by general readers from various sectors as microbiology and omics next to environmental sciences.

Minor comments:
- correct in abstract: “genome. Synteny”
- style of references need to be streamlined in the text
- Line 146 to 150; the sentence could be rephrased to be better readable.
- Line 213: “A.” italics style
Line 222: “P450” correct
Line 302: “ phage regions … “ correct
Fig 5. Needs some improvement as the network is hard to figure out; maybe the color code and designations can be optimized.
- Suppl. Material files (both) are needed for the manuscript. They are fine.

Reviewer 2 ·

Basic reporting

- The english is poor. There are many serious errors in grammar and spelling. Many sentences are difficult to understand in the manuscript. Some corrections are given below, but the text should be checked again.

- References and background are sufficient.

- The article structure is professional and raw data is shared.

- Results are relevant.

Experimental design

- The research is within the scope of the journal.

- The research question is well defined and a knowledge gap is filled (a new strain is sequenced).

- The research was well conducted, but some important details are missing :

1) Line 121-122. What is the name of the reference sequence? Check the accession number given in the text (no results were found on NCBI).

2) Figure 1 : it is said in the text, line 159, that 2049 contigs were obtained. But Fig 1 presents one circular chromosome. Was the genome of strain BHW-15 closed or not? Is it the reference with the 2049 contigs mapped on it? If yes, were all the gaps filled? This is not clear and this point must be clarified. Moreover, in Figure 1, the colors for the prophage types (red, gray and green) are not visible on the circular chromosome.

Validity of the findings

- The discussion concludes that strain is metal-resistant because 35 metal resistance genes were detected in the genome. However, genes are not always expressed or functional and no data is provided concerning the Minimal Inhibitory Concentration for metals in the strain (Cu, Zn, As, Cr, etc.). Before concluding that strain BHW-15 is metal resistant the strain should be cultivated on media with different concentrations of metals, and compared to control strains cultivated on the same media. Alternatively, resistance proteins may be detected by proteomics.

Additional comments

The manuscript is interesting and should be published as it describes the genome of a new bacterial strain. However, in its present state it is not clear in the manuscript if the genome was completely closed or not. In addition, the metal resistance phenotype of the strain (MIC) should be determined before publication (antibiotic resistance was determined, but not metal resistance). Finally, the english should be revised and some corrections are given below.

Minor points :
- Replace "Achromobactor" by "Achromobacter" in the title of the manuscript.
- Introduction, line 55. Replace "it" by "them".
- Line 57 : bad english, rephrase.
- Line 59. "Heavy metals" is a meaningless term. The term "metals" should be used in the text (Duffus et al. 2002. Pure Appl. Chem., Vol. 74, No. 5, pp. 793–807).
- Line 60 : replace "maintain" by "maintains".
- Line 65, bad english. Remove the word "rather" and rephrase.
- Line 70 : replace "are directly polluting the water" by "may contaminate water".
- Lines 79-82. These two sentences are hard to understand. What are the "non-antibiotic agents"?? Please rephrase and explain.
- Line 98. "l" should be replaced by "L" everywhere in the text.
- Line 103 : replace "of brown" by "of a brown".
- Line 103 : replace "harborage of" by "the presence of an".
- Line 107 : to confirm the presence of what?
- Line 144 : What medium was used? What was the size of the discs?
- Line 149 : replace "inferring" by "referring".
- Line 158 : replace " xylosoxidnas" by "xylosoxidans".
- Line 161 : What is "In RAST by RASTtk"? Have you used RAST or RASTtk? Please give a reference for RASTtk.
- Line 176 : replace "arsenic oxidizing" by "arsenite oxidizing".
- Line 177 : replace "arsenic reducing" by "arsenate reducing".
- Line 182 : replace "transporter phosphate" by "phosphate transporter".
- Line 194 : "distiction". Do you mean "distinction"? Also remove the word "high".
- Line 201 : replace "In vitro..." by "Using in vitro..."; Replace "test" by "tests".
- Line 205 : replace "contains" by "contained".
- Line 208 : replace "this" by "these".
- Line 220 : replace "Terpinoids" by "terpenoids".
- Line 251-252 : what are "other high arsenic converting bacteria"? Do you mean other bacteria able to metabolize arsenite and arsenate?
- Line 255 : replace "genomic possession" by "genes".
- Line 267 : Tn21 : please give more details on that transposon (a drawing?)
- Line 305 : replace "facilitated" by "facilitate".
- Line 306 : replace "bacteria" by "bacterium".
- Line 307 : what do you mean by "this two resistance"?
- Line 325 : "It can inferred"... ? Rephrase. Do you mean "It can be inferred"?
- Line 326 : "to thrive in ease"?


- Table 2 : replace "aiS" by "aioS".
- Figure 4 (A) : explain in the legend why two red arrows are represented for aioA in strain BHW-15 (two aioA genes in different overlapping reading frames?).

Reviewer 3 ·

Basic reporting

1.1. The English phrasing, grammar and language should be improved to make sure that the audience understand the message. I suggest rephrasing lines 56-57, 201-202, 217-224, 283-285, 322-325. I suggest paying attention to grammar mistakes such as verb and subject agreement (for example, lines 115, 126,215, 252), missing “s” (ex. lines 181, 236, 248, 252, 258, 299) and italic words (for example, in vitro). I suggest reconsidering the use of the word “heavy” for heavy metals as this notion is controversial.
1.2. I would recommend the authors to improve the field background by (i) explaining why they investigated groundwaters and bringing relevant information concerning metal and antibiotic contamination of this kind of environment. (ii) reference this background with more recent and appropriated literature.
1.3. Figures should be improved to display sufficient resolution to permit the audience to read text on the figure. It includes Figure 1, Figure 3. Besides, Supplemental file 1 contains a table with to column that are difficult to understand and appreciate due to its untidy structure. Supplemental File 2 is titled “Supplemental file 4”.
1.4. Parts of the discussion lack references such as in lines 297-299, 299-305 or 333-336.

Experimental design

2.1. For a better answer to the research question, I would recommend authors to provide information from the literature or determine in the present study the characteristics of waters from which the strain was isolated, including metal and antibiotic contents.
2.2. I would suggest providing more information about the growth medium, KMnO3 and AgNo3 screening principles to permit the audience to perform that kind of analysis (line 96)
2.3. No information is provided concerning the network analysis network (Figure 5) nor adequate discussion concerning these results.
2.4. The research question should be restructured to make sure that the audience understand the aim of the study. In other words, I would insist to say that this study aims to genomic basis to understand antibiotic and metal cross-resistance in Achromobactor xylosoxidans BHW-15 in further experiments.

Validity of the findings

3.1. Discussion should be enriched and explore links between the other strains of Achromobactor xylosoxidans and its metal and antibiotic resistance potential. I would recommend, for instance,
Jakobsen, T.H., Hansen, M.A., Jensen, P.Ø., Hansen, L., Riber, L., Cockburn, A., et al. (2013) Complete Genome Sequence of the Cystic Fibrosis Pathogen Achromobacter xylosoxidans NH44784-1996 Complies with Important Pathogenic Phenotypes. PLoS One 8: 8–11.
Amoureux, L., Bador, J., Fardeheb, S., Mabille, C., Couchot, C., Massip, C., et al. (2013) Detection of Achromobacter xylosoxidans in hospital, domestic, and outdoor environmental samples and comparison with human clinical isolates. Appl. Environ. Microbiol. 79: 7142–7149.
3.2. Linked with the research question, authors should bring clues to complete these results and decipher cross-resistance mechanisms

Additional comments

The study provides relevant information to decipher the link between metal and antibiotic resistance. It provides robust sequencing raw data and relevant analysis of the genome of Achromobactor xylosoxidans BHW-15. If there are weaknesses, these are the technically incorrect English as mentioned above, the insufficient resolution of figures, the lack of references in the discussion and information concerning antibiotic and metal contents in groundwater ecosystems
I would also reorganize the discussion to start with general findings and end with specific findings.

---

## Round 0.2 · accepted · Accept

Thank you for your efforts in revising your manuscript.

# Reviewer 1 ·

Basic reporting

The revised manuscript improved significantly with respect to language and presentation. A proper number of references have been used and as far as I can judge these cover the field of research in a good way. The article is well prepared and structured thus specialist and general reader can easily follow and take the messages in a good order.

Thus I have no reservation about the style and content.

Experimental design

The design of experiments is appropriate for this study aiming to investigate the drawn hypothesis of As contaminated sites and microbial responses on differnet organisation levels.

Methods are detailed described and I have the feeling they can be reproduced.

Validity of the findings

As mentioned above the work is solid!

The results are clear and easy to follow and it adds novelty to the field.

The experiments were proper to gain these results and this allowed to draw a number of conclusions - to my optinion the correct one have been suggested and stated. Here additional references were discussed.

It seems appropriate in terms of length and content.

Additional comments

The authors improved the manuscript quite a lot and the presentation is now more clear. It will be interesting to follow this topic as I think there is room to identify more As resistance correlations - it would be interesting to even study more in detail the antibiotica resistance and occurence of (heavy) metal realted issues.